# Hypothermia after Perinatal Asphyxia Does Not Affect Genes Responsible for Amyloid Production in Neonatal Peripheral Lymphocytes

**DOI:** 10.3390/jcm11123263

**Published:** 2022-06-07

**Authors:** Agata Tarkowska, Wanda Furmaga-Jabłońska, Jacek Bogucki, Janusz Kocki, Ryszard Pluta

**Affiliations:** 1Department of Neonate and Infant Pathology, Medical University of Lublin, 20-093 Lublin, Poland; agatatarkowska@umlub.pl (A.T.); wm.jablonska@gmail.com (W.F.-J.); 2Department of Organic Chemistry, Faculty of Pharmacy, Medical University of Lublin, 20-093 Lublin, Poland; jacekbogucki@wp.pl; 3Department of Clinical Genetics, Medical University of Lublin, 20-080 Lublin, Poland; januszkocki@umlub.pl; 4Laboratory of Ischemic and Neurodegenerative Brain Research, Mossakowski Medical Research Institute, Polish Academy of Sciences, 02-106 Warsaw, Poland

**Keywords:** perinatal asphyxia, hypoxic-ischemic encephalopathy, hypothermia, brain ischemia, Alzheimer’s disease, lymphocytes, amyloid protein precursor, β-secretase, presenilin 1 and 2, genes

## Abstract

In this study, the expression of the genes of the *amyloid protein precursor*, *β-secretase*, *presenilin 1* and *2* by RT-PCR in the lymphocytes of newborns after perinatal asphyxia and perinatal asphyxia treated with hypothermia was analyzed at the age of 15–21 days. The relative quantification of Alzheimer’s-disease-related genes was first performed by comparing the peripheral lymphocytes of non-asphyxia control versus those with asphyxia or asphyxia with hypothermia. In the newborns who had perinatal asphyxia, the peripheral lymphocytes presented a decreased expression of the *amyloid protein precursor* and *β-secretase* genes. On the other hand, the expression of the *presenilin 1* and *2* genes increased in the studied group. The expression of the studied genes in the asphyxia group treated with hypothermia had an identical pattern of changes that were not statistically significant to the asphyxia group. This suggests that the expression of the genes involved in the metabolism of the amyloid protein precursor in the peripheral lymphocytes may be a biomarker of progressive pathological processes in the brain after asphyxia that are not affected by hypothermia. These are the first data in the world showing the role of hypothermia in the gene changes associated with Alzheimer’s disease in the peripheral lymphocytes of newborns after asphyxia.

## 1. Introduction

According to existing data, 20 out of 1000 live births require resuscitation, resulting in clinical and biochemical symptoms of perinatal asphyxia [1,2]. Currently, perinatal asphyxia affects approximately 4 million newborns annually worldwide [3]. About 15–25% of these newborns die in the neonatal period, up to 25% of survivors develop neurological deficits, and 10–30% have retarded psychomotor development [4,5]. Studies analyzing the long-term consequences in preschool children after perinatal asphyxia have shown that cerebral palsy develops in 5.5–52% [6,7,8,9], severe disability in 11–19% [6], various motor impairments in 1.3–40% [2,8,9,10], hearing loss in 2–20% [6,9,10], vision defects in 1.8–40% [6,9,11], speech disorders in 4.2–21% [6,7,8], epilepsy in 13%, and cognitive impairment in 31% [1,2,12].

There is increasing evidence that the pathology induced in the brains of newborns during and after perinatal asphyxia is similar to that found in adult neurodegenerative diseases such as Alzheimer’s disease [3,13,14,15]. Experimental studies have shown that perinatal asphyxia in mice causes delayed neuronal death in the hippocampus and significant deficits in memory and spatial learning [13,14]. In addition, significantly higher levels of tau protein and its increased phosphorylation, higher concentrations of amyloid protein precursor, decreased hypoxia-inducible factor 1 alpha, lower levels of amyloid-degrading neprilysin, an increased accumulation of amyloid with the activation of microglia, and astrocytes in the brain after asphyxia were found [3,13,15,16]. Furthermore, one study found elevated blood levels of tau protein after asphyxia on postnatal days 3 and 7 [17]. Another study showed a significant decrease in β-amyloid peptide 1–42 in the cerebrospinal fluid of neonatal pigs after perinatal asphyxia [18]. Based on the amyloid hypothesis of Alzheimer’s disease, a reduction in the β-amyloid peptide 1–42 in the cerebrospinal fluid is considered to be the first hallmark in the development of Alzheimer’s disease [19]. In the prodromal and preclinical stages of Alzheimer’s disease, the level of β-amyloid peptide 1–42 in the cerebrospinal fluid is reduced [20]; a similar pattern was noted in experimental perinatal asphyxia [18].

In our previous study, we assessed the expression of genes related to the development of Alzheimer’s disease in the peripheral lymphocytes of newborns after perinatal asphyxia [3]. Following perinatal asphyxia, we found a decreased expression of *HIF 1*-*α*, *amyloid protein precursor*, and *β-secretase* genes in neonatal lymphocytes [3]. However, we observed a significant overexpression of the *presenilin 1* and *2* genes related to γ-secretase in lymphocytes 15 days or more after perinatal asphyxia [3].

Currently, hypothermia remains the only available treatment option for neonates after perinatal asphyxia with variable and/or transient medical benefits [5,21,22,23]. Some studies show that treating perinatal asphyxia with hypothermia improves survival and reduces the incidence of long-term complications, such as severe disability, epilepsy, and the severity of cerebral palsy [24,25]. By contrast, other studies have shown that treatment with hypothermia reduces the incidence of death or severe impairment at 18 months of survival by 33% in children with a history of perinatal asphyxia, and does not affect adverse complications after asphyxia, such as neurodevelopmental disorders and cerebral palsy [6,21,26]. Further studies showed that in children aged 6–8 years, after hypothermia treatment, the results of cognitive assessment were significantly lower compared to healthy children [27,28]. Both preclinical and clinical studies show that treatment with hypothermia does not reduce the problems associated with memory impairment [29,30]. Unfortunately, this method of treating children with perinatal asphyxia, if successful, is only temporary or incomplete [5,6,23,31]. The above is confirmed by data showing that about 10% of infants with perinatal asphyxia treated with hypothermia have epilepsy, 20% have cerebral palsy [5,6], and about half die [31] or develop severe functional deficits [27,28].

Therefore, the studies conducted in the last ten years have shown the transient efficacy of treatment with hypothermia in the early stages, but not in all children with a history of perinatal asphyxia. Furthermore, in the late stages after asphyxia, this treatment has shown a significant lack of impact on pathological changes, especially in cognitive function [27,28], which is associated with the development of brain neurodegeneration. In a previous qPCR study, we found that perinatal asphyxia influences the downregulation of the *amyloid protein precursor* and *β-secretase* and the upregulation of the *presenilin 1* and *2* genes in a group of survivors of 15 days or more (Figure 1 and Figure 2) [3]. We then selected these genes for further studies after the routine treatment with hypothermia and showed that there is no difference in the expression of the above genes in peripheral lymphocytes in treated and untreated patients. The studies show that changes in the expression of the genes related to the metabolism of the amyloid protein precursor, resulting from asphyxia, persist in lymphocytes after hypothermia therapy.

## 2. Materials and Methods

### 2.1. Study Setting and Design

In this study, we compare children with perinatal asphyxia with a group of children with perinatal asphyxia treated with hypothermia with survival exceeding 15 days, born in the Lublin region, Poland. During the study, all newborns were hospitalized at the Department of Neonate and Infant Pathology at the Medical University of Lublin. The inclusion criteria for asphyxia were defined as follows:Newborns (full-term and preterm) > 31 weeks of gestational age.Metabolic acidosis with pH < 7.0 (in umbilical cord or newborn blood sample obtained during 60 min after birth),or Base deficit > −12,or Apgar score of 0–5 at 10 min or continued need for resuscitation at 10 min of age.Presence of multiple organ-system failures,Clinical evidence of encephalopathy: periodic breathing/apnea, abnormal oculomotor or pupillary movements, weak or absent suck, or clinical seizures.Neurologic findings cannot be attributed to other cause.

The investigative procedure was approved by the Ethics Committee of the Medical University of Lublin (25 April 2013, consent no. KE-0254/118/2013). After obtaining the written consent of all authorized caregivers of patients in both study groups, blood was collected along with blood sampling for regular laboratory analysis for diagnosis and/or treatment.

### 2.2. Therapeutic Protocol for Hypothermia

The Children’s Hospital of the Medical University of Lublin is the only such hospital in Lublin voivodship that treats newborns after perinatal asphyxia with hypothermia. Therefore, all newborns born with features of severe or moderate perinatal asphyxia in this voivodship are transported there for treatment by hypothermia. During transport, passive cooling begins (the heat radiator in the incubator is off) under the control of the rectal temperature. As there is only one hypothermia treatment center in a large area, adjusting to the treatment window (<6 h) is a major challenge. Thus, for treatment by hypothermia, newborns must be admitted to the intensive care unit within 6 h of birth. After passing the criteria for inclusion in treatment, as outlined below, the cooling procedure begins. The current Polish qualification protocol for inclusion or exclusion for head- or whole-body cooling is presented below.

(1)Inclusion criteria for perinatal asphyxia treated by hypothermia:
▪Newborns born after ≥35 weeks of gestation with severe or moderate perinatal asphyxia,▪or Apgar ≤ 5 point at 10th minute of life,▪or mechanical ventilation 10 min after birth,▪or acidosis pH < 7.0 in umbilical or arterial blood 1 h after birth,▪or base deficit of at least −16 mmol/L in umbilical or arterial blood 1 h after birth.▪or Neurological deficits: coma, lethargy, decreased muscle tone, abnormal response to stimuli, no or reduced sucking reflex, convulsions, increased muscle tone.▪Changes in amplitude—integrated EEG recordings.(2)Exclusion criteria:
▪Gestational age < 35 weeks.▪Birth weight < 1800 g.▪Severe birth defects with poor prognosis.▪Severe traumatic head injury.▪Rectal obstruction which prevents deep body temperature measurements.▪Lack of consent of parents/legal guardians.▪Over 6 h of life on admission to the intensive care unit.

In our study, patients not treated with cooling (the non-hypothermia group) were excluded from treatment due to their gestational age, lack of parental consent, or excessive amount of time from birth to cooling center (>6 h). All hypothermic patients (group) received whole-body cooling using the CritiCool device (Belmont Medical Technologies, Billerica, MA, USA), according to standard guidelines for newborns in Poland. After starting and calibrating the CriticCool, the naked newborns were placed in the cooling cover. The surface-temperature probe was placed on the chest and the rectal-temperature probe at a depth of 5 cm. The target body temperature was 33.5 °C (rectal temperature between 33.0 °C and 34.0 °C). During hypothermia, all infants were monitored and their rectal temperature, heart rate, respiratory rate, pulse oximetry, blood pressure, and real-time skin temperature were recorded continuously. Amplitude-integrated EEG records were visible simultaneously with the patient’s vital signs. General management of the infants was in accordance with routine clinical practice. After 72 h of cooling down, heating was initiated with the same device. The temperature was increased by 0.1 °C every 20 min, which gave an overall temperature increase of 0.3 °C over 1 h. It took about 10 h for the babies to warm to normal temperature. The rate and duration of the heating were individually adjusted, depending on the response and clinical condition of the infant. The rectal temperature was monitored for at least 24 h after reheating. In the studied patients, MRI of the brain was usually performed 3–7 days after heating. Transcranial ultrasound was performed in the first days of life and repeated after hypothermia, depending on clinical indications.

### 2.3. Study Population and Sample Size

The study included 20 neonates, 5 newborns who had perinatal asphyxia, plus 5 healthy neonates as controls, 5 newborns who had asphyxia with hypothermic treatment, and 5 healthy neonates as a control. The test and control groups were 15–21 days old. In the asphyxia group, there were 3 girls aged 21, 15, and 21 days and 2 boys aged 21 and 18 days. In the asphyxia group with hypothermia, there were 3 girls aged 15, 21, and 15 days and 2 boys aged 21 days. There were 5 newborns in the control groups, who were of the same age and sex as in the study groups. The control children were hospitalized with infant colic, breast-milk jaundice, mild respiratory infections, and poor weight gain. Newborns from the control groups were born after uncomplicated pregnancies and without any health problems. Newborns from these groups were hospitalized at the Department of Neonate and Infant Pathology of the Medical University of Lublin for diagnosis and treatment due to the symptoms presented above. Clinical characteristics of all enrolled neonates are presented in Table 1.

### 2.4. Study of Gene Expression, Including Amyloid Protein Precursor, β-Secretase, and Presenilin 1 and 2

Venous blood was collected for citrate for gene studies in lymphocytes. Next, the blood was placed in a centrifugal tube (Falcon, Coming Science Mexico S.A. de C.V., Reynosa, Tamaulipas, Mexico) and diluted 1:1 with PBS (Biomed, Lublin, Poland). The blood prepared in this way was combined with the Gradisol L reagent (Polfa, Lublin, Poland). It was then centrifuged in a Centrifuge 5810R (Eppendorf, Hamburg, Germany) at room temperature at 2000 rpm for 20 min. After the blood mononuclear cells were gathered and washed in 5 mL PBS (Biomed, Poland), they were centrifuged at room temperature at 2000 rpm for 10 min. The cells were then placed in 1 mL PBS (Biomed, Poland) and centrifuged again in a MiniSpin plus centrifuge (Eppendorf) at room temperature at 2000 rpm for 10 min. RNA was then isolated from the pellet. RNA isolation was performed according to the previously prepared procedure [32,33]. After adding cooled TRI (4 °C) (Invitrogen, Thermo Fisher Scientific’s, Warsaw, Poland) to the sediment at a volume of 0.5 mL, the material was homogenized manually. Next, 0.2 mL of chloroform was added to the samples, which were shaken for 15 s and then left at room temperature for 15 min. All samples were then centrifuged for 15 min in a 5415 R centrifuge at 4 °C and 13,600 rpm and total cellular RNA was precipitated from the aqueous phase by adding 0.5 mL of isopropyl alcohol. In the last phase, all samples were centrifuged for 20 min in a 5415R centrifuge at 4 °C at a speed of 13,600 rpm. A Nano Drop 2000 spectrophotometer (Thermo Fisher Scientific, Waltham, MA, USA) was used to evaluate the quality and quantity of RNA. The obtained RNA in 80% ethanol and the temperature of −20 °C was stored for further research. One µg of RNA was used to transcribe cDNA using the cDNA reverse transcription kit according to the manufacturer’s instructions (Applied Biosystems, Foster City, CA, USA). A cDNA synthesis was performed on a Veriti Dx (Applied Biosystems, Foster City, CA, USA) in the following order: phase I 25 °C, 10 min; phase II 37 °C, 120 min; phase III 85 °C, 5 min; phase IV 4 °C. The cDNA obtained by this method was amplified by real-time gene expression analysis (PCR) on a 7900HT Real-Time Fast System apparatus (Applied Biosystems, Foster City, CA, USA) with the Master Mix SYBRgreen PCR power mix reagent, using the manufacturer’s SDS software (730 System Software, Applied Biosystems, Waltham, MA, USA) [32,33]. The production of multiple copies of the DNA sequence began with denaturation at 95 °C for 10 min and was followed by 40 cycles each at two different temperatures: 95 °C, for 15 s, and 60 °C, for 1 min. For each amplification cycle, checking and calculation of the amount of DNA reproduction was performed on a 7900HT Real-Time Fast System (Applied Biosystems, Foster City, CA, USA). Test software (Applied Biosystems, Foster City, CA, USA) was used to estimate the number of PCR cycles for which the level of fluorescence exceeded the specified expression threshold cycle (CT) in order to calculate the number of DNA molecules in the mixture at the start of the reaction. Normalization was achieved against the endogenous glyceraldehyde 3-phosphate dehydrogenase control gene. The relative quantification (RQ) of the gene expression was assessed based on the DCT technique, and the values were calculated as RQ = 2^−^^ΔΔCT^ [32,33]. The RQ calculation of the tested genes was performed using Expression Suite Software v.1.1 (Thermo Fisher Scientific, Waltham, MA, USA). The RQ results were finally obtained after logarithmic RQ (LogRQ) conversion [32,33]. LogRQ = 0 indicates that the changes in gene expression do not differ between experiments and controls. LogRQ < 0 indicates that gene expression is decreased in the asphyxia study. LogRQ > 0 indicates increased gene expression in the asphyxia study compared to the control. The following TaqMan probe sets (Applied Biosystems, Foster City, CA, USA), labeled FAMNFQ, were used: Hs00169098_m1 (*APP*), Hs01121195_m1 (*BACE1*), Hs00997789_m1 (*PSEN1*), Hs01577197_m1 (*PSEN2*), and Hs99999905_m1 (*GAPDH*). Statistical analysis of the results was performed with Statistica v. 13.3 software (Tibco Corporation, Palo Alto, CA, USA) using the Mann-Whitney U test. The results are presented as mean ± SD and statistically significant when *p* ≤ 0.05.

## 3. Results

### 3.1. Expression of Genes Metabolizing Amyloid Protein Precursor in Lymphocytes after Asphyxia

After perinatal asphyxia, the average expression level of the *amyloid protein precursor* gene in the study group was below that of the control values (Figure 3). The minimum change was −0.370-fold, the maximum change was −0.039-fold, and the median change was −0.317-fold. The mean value of the expression of the *β-secretase* gene was −0.446, the minimum change was −0.800-fold, the maximum change was −0.079-fold, and the median was −0.428-fold. The mean value of the expression of the *presenilin 1* gene was 0.339, the minimum change was 0.059-fold, the maximum change was 0.658-fold, and the median change was 0.360-fold. The mean value of the expression of the *presenilin 2* gene was 0.240, the minimum change was 0.184-fold, the maximum change was 0.351-fold, and the median change was 0.195-fold. Figure 3 illustrates the changes in the average expression level of the studied gene in the lymphocytes after asphyxia.

### 3.2. Expression of Genes Metabolizing Amyloid Protein Precursor in Lymphocytes after Asphyxia Treated with Hypothermia

After hypothermia-treated perinatal asphyxia, the mean expression level of the *amyloid protein precursor* gene in the study group was below that of the control (Figure 3). The minimum change was −1.347-fold, the maximum change was −0.038-fold, and the median change was −0.523-fold. The mean value of the *β-secretase* expression gene was −0.662, the minimum change was −1.409-fold, the maximum change was −0.088-fold, and the median change was −0.512-fold. The mean value of the *presenilin 1* gene expression was 0.491, the minimum change was 0.196-fold, the maximum change was 0.993-fold, and the median change was 0.476-fold. The mean value of the expression of the *presenilin 2* gene was 0.568, the minimum change was 0.140-fold, the maximum change was 0.867-fold, and the median change was 0.626-fold. Figure 3 illustrates the changes in the mean expression level of the tested gene in the lymphocytes after asphyxia treatment with hypothermia. There were no significant statistical changes between the asphyxia group and asphyxia with hypothermia treatment (Figure 3).

## 4. Discussion

The present study was performed to determine whether the peripheral blood lymphocytes in patients with perinatal asphyxia treated with hypothermia show changes in the expression of the genes associated with amyloid generation that are simultaneously involved in the response to asphyxia. This work is an extension of current research [3], in which we present new changes in the genes related to Alzheimer’s disease, such as the *amyloid protein precursor*, *β-secretase*, *presenilin 1*, and *presenilin 2*, in the circulating lymphocytes of neonates with perinatal asphyxia treated with hypothermia. It is highly probable that the increase in the expression of the *presenilin 1* and *2* genes starting 15 days after perinatal asphyxia reflects the onset of the progressive neurodegeneration of the brain as a response to asphyxia [14,29], while pointing to the systemic nature of the perinatal and postnatal asphyxia injury. This suggestion is also supported by a significant increase in the level of tau protein in the brain, as investigated by microdialysis in hypothermic global cerebral ischemia [16]. The lack of influence of hypothermia on the genes metabolizing the amyloid protein precursor in peripheral lymphocytes after perinatal asphyxia affects the function and activation of lymphocytes.

Hypoxia is known to alter the function of lymphocytes [34,35] and is involved in the migration and activation of lymphocytes in the brain after asphyxia, ultimately contributing to brain damage in newborns [36,37]. Lymphocytes are at risk of hypoxia in the circulatory system during perinatal asphyxia, but they are also at risk of hypoxia when they leave the bloodstream and enter hypoxic brain tissue. Thus, in perinatal asphyxia, where the entire brain is hypoxic, leukocytes are additionally exposed to hypoxia in the brain tissue. Evidently, the dual influence of hypoxia on lymphocytes may favor the progression of neurodegeneration [37].

Ultrastructural studies of brains following experimental ischemia revealed numerous clusters of platelets of various sizes within both arterial and venous interstitial vessels [38]. Platelets at different disintegration stages have also been observed. The platelets adjacent to endothelial cells were often degranulated, with shape changes, including pseudopodia. Platelet aggregates were random, focal, and more prevalent in the brain cortex, basal ganglia, thalamus, hippocampus, and cerebellum [38]. Recent aggregates of platelets and thrombi have also been found, showing varying degrees of degranulation. Platelets were also found outside the cerebral vessels in brain tissue [38]. Evidence was provided that platelet aggregation recurred long after the ischemic event, i.e., one year [38]. The local accumulation of platelets in microvessel branches or vascular bifurcation areas was predominant [38], which correlated well with changes in the blood–brain barrier [39]. The number of platelet aggregates increased with longer post-ischemic recirculation times. The local adhesion of platelets to endothelial cells was frequently noted post-ischemia [38]. The accumulation of platelets during vascular thrombosis causes the abundant release of amyloid, indicating an additional source of amyloid in the brains of Alzheimer’s disease patients that may emerge in the event of frequent micro-thrombotic episodes in these patients [40]. Based on a literature review, it appears that platelet-derived amyloidosis is now more common than previously thought [41]. Thus, systemic amyloids derived from platelets may be involved in a variety of diseases with progressive amyloidosis, ranging from brain cancer, pre-eclampsia, and skin amyloid accumulation to Alzheimer’s disease [41]. This partly suggests the chronic activation of platelets in Alzheimer’s disease [42]. This in turn suggests that a similar phenomenon may also contribute to neurodegeneration in perinatal asphyxia.

The proteins related to neurodegeneration have been shown to be present in human lymphocytes and to interact with proteins related to inflammation and autophagy [43]. Lymphocytes penetrate the brain at a late stage after cerebral ischemia and are involved in the progression of neuroinflammation [36]. The increased expression of the presenilin genes related to γ-secretase metabolizing the amyloid protein precursor to amyloid positively correlates with the staining of the presenilin and amyloid in lymphocytes in neurodegenerative diseases [43]. The analysis of immunopeptide databases provides evidence that the neurodegeneration proteins present in lymphocytes are a source of endogenous neurotoxic proteins in the brain, such as the amyloid protein precursor, presenilin 1 and 2, and tau protein [43]. There are also studies showing that presenilins affect calcium levels, amyloid protein precursor transport, immune and mitochondrial function, and the expression and activity of β-secretase in the brain [44,45]. Presenilins are elevated in Alzheimer’s disease and increase the likelihood of its development [45,46]. It has been shown that the activity of presenilins can be divided into γ-secretase-dependent and γ-secretase-independent [47]. An example of the γ-secretase-dependent action of presenilins is the presence of cuts during the metabolism of the amyloid protein precursor [47]. Moreover, the γ-secretase-independent action of presenilins includes the stabilization of β-catenin in the Wnt signaling pathway, the regulation of calcium homeostasis, and their interaction with synaptic transmission [47].

New evidence suggests that pre- and post-ischemic stroke processes in humans may trigger amyloid acumulation in the subcortical area and hippocampus, supporting the notion that amyloid deposition begins early in Alzheimer’s disease [48,49]. Furthermore, the immunoreactivity of apolipoprotein E in neuronal cells was also significantly increased in ischemic CA1 and CA3 pyramidal neurons of the hippocampus [48]. These data strongly support the commonly observed link between vascular risk factors and/or brain ischemia epizodes and the progression of Alzheimer’s disease [49].

Changes in the expression of the studied genes in lymphocytes are highly likely to influence the development or intensification of pathological pathways in the hypoxic brain. In particular, upregulated presenilin genes are involved in γ-secretase, which participates in amyloid production. We suggest that changes in lymphocytes could potentially affect the production of amyloid in these cells and, indirectly, in the peripheral circulation and in the brain tissue after perinatal asphyxia. This is a new observation, which suggests that in both untreated newborns and in those treated with hypothermia, perinatal asphyxia can alter the function of circulating lymphocytes by upregulating γ-secretase-related genes.

## 5. Conclusions

The question underlying this study was whether the expression of genes related to the metabolism of the amyloid protein precursor in the peripheral lymphocytes of newborns after perinatal asphyxia could be modified by hypothermia. It was noted that of the neonates who had perinatal asphyxia, both those who were untreated and those who were treated with hypothermia showed identical changes in gene expression related to the processing of the amyloid protein precursor in the lymphocytes (Figure 3). Taking into account the passage of lymphocytes after perinatal asphyxia to the brain, our observations indicated a possible role of lymphocytes in progressive neurodegenerative processes, despite the treatment of the newborns who had perinatal asphyxia with hypothermia. The activation and expression of γ-secretase-related *presenilin* genes in these cells suggests their potential involvement in the production of amyloid in the brain after asphyxia. These findings could lead to future advanced research to understand the exact role of late-stage lymphocyte responses after perinatal asphyxia. In perinatal and postnatal asphyxia, blood can be used for a thorough examination of the course of asphyxia. The main limitation of this article is the relatively small number of newborns in the study and control groups. This meant that there was not enough material to extend the Western-blot protein research. This study should be treated as a preliminary investigation, which should be continued on a larger group of newborns.

## Figures and Tables

**Figure 1 jcm-11-03263-f001:**
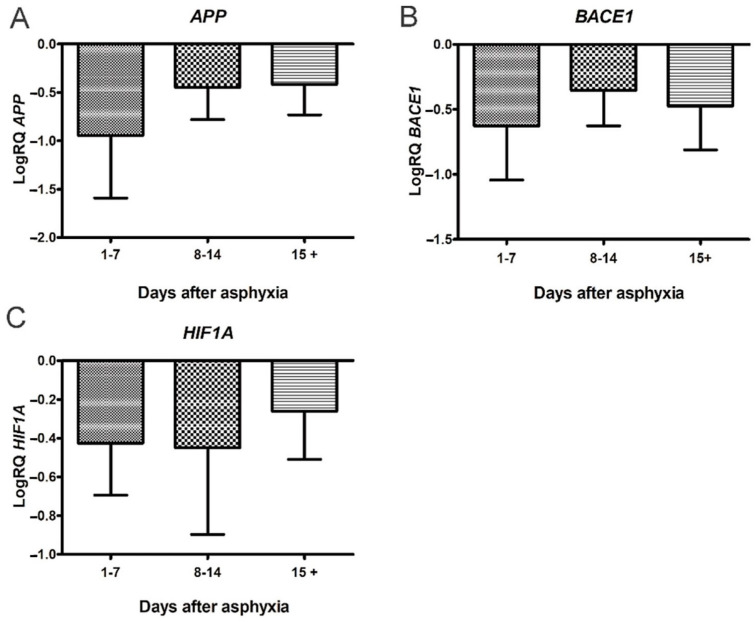
Changes of *amyloid protein precursor (APP), β-secretase* (*BACE1)* and *hypoxia-inducible factor 1-α* (*HIF1A*) genes after perinatal asphyxia. (**A**) Mean levels of *APP* gene expression in lymphocytes after perinatal asphyxia in age groups 1–7 (n = 8), 8–14 (n = 7), and 15+ (n = 8) days. Marked SD, standard deviation. N, number of children in the group. The changes between the groups were not statistically significant (Kruskal–Wallis test). (**B**) Mean levels of *BACE1* gene expression in lymphocytes after perinatal asphyxia in age groups 1–7 (n = 8), 8–14 (n = 7), and 15+ (n = 8) days. Marked SD, standard deviation. N, number of children in the group. The changes between the groups were not statistically significant (Kruskal-Wallis test). (**C**). Mean levels of *HIF1A* gene expression in lymphocytes after perinatal asphyxia in age groups 1–7 (n = 8), 8–14 (n = 7), and 15+ (n = 9) days. Marked SD, standard deviation. N, number of children in the group. The changes between the groups were not statistically significant (Kruskal-Wallis test) [3].

**Figure 2 jcm-11-03263-f002:**
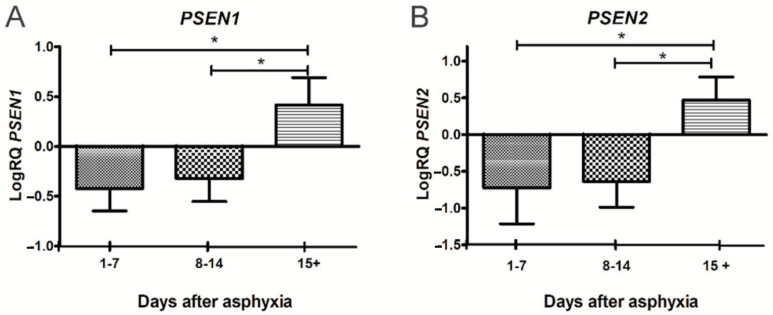
Changes of *presenilin 1 (PSEN1)* and *presenilin 2 (PSEN2)* genes after perinatal asphyxia. (**A**). Mean levels of *PSEN1* gene expression in lymphocytes after perinatal asphyxia in age groups 1–7 (n = 8), 8–14 (n = 7), and 15+ (n = 10) days. Marked SD, standard deviation. N, number of children in the group. The changes between the 1–7 and 8–14 day groups were not statistically significant. The indicated statistically significant differences in the level of gene expression between groups 1–7 and 15+ days (z = 3.903, *p* = 0.0002) and between 8–14 and 15+ days (z = 3.092, *p* = 0.0059) after perinatal asphyxia (Kruskal-Wallis test). * *p* ≤ 0.01. (**B**). Mean levels of *PSEN2* gene expression in lymphocytes after perinatal asphyxia in age groups 1–7 (n = 8), 8–14 (n = 7), and 15+ (n = 10) days. Marked SD, standard deviation. N, number of children in the group. The changes between the 1–7 and 8–14 day groups were not statistically significant. The indicated statistically significant differences in the level of gene expression between groups 1–7 and 15+ days (z = 3.634, *p* = 0.0008) and between 8–14 and 15+ days (z = 3.387, *p* = 0.002) after perinatal asphyxia (Kruskal-Wallis Test). * *p* ≤ 0.01. [3].

**Figure 3 jcm-11-03263-f003:**
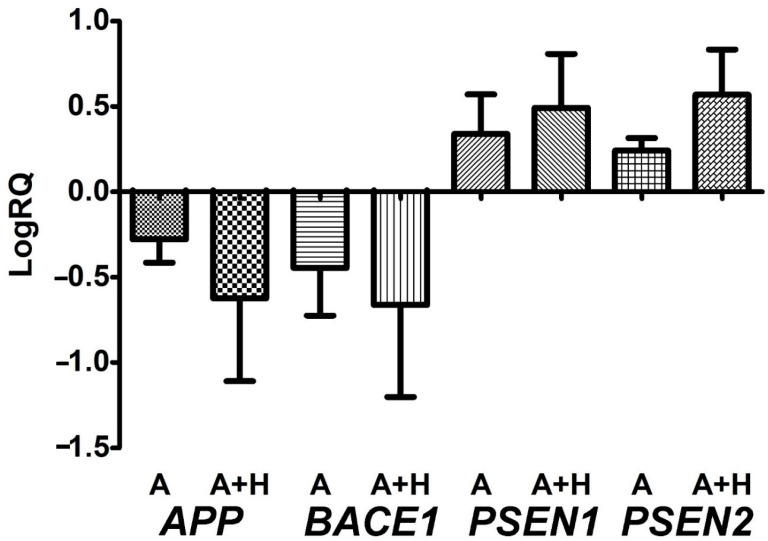
Mean levels of *amyloid protein precursor (APP*), *β-secretase* (*BACE1*), *presenilin 1* (*PSEN1*), and *presenilin 2* (*PSEN2*) gene expression in peripheral lymphocytes after perinatal asphyxia (A) and perinatal asphyxia with hypothermia (A + H) with survival of more than 15 days. Marked SD, standard deviation. The changes between the groups were not statistically significant (Mann-Whitney test).

**Table 1 jcm-11-03263-t001:** Clinical characteristics of enrolled neonates.

Group	Age (Days)	Gestationl Age (Weeks)	Birth Weight (g)	Apgar Score (1 min)	RBC (×1000/μL)	WBC (/μL)	Lymphocyte (/μL)	PLT (×1000/μL)	Hct (%)	pH	BE (Mmol/L)
**Control**	20 ± 1	37 ± 2	3274 ± 542	10 ± 0	4204 ± 416	11,816 ± 1692	6256 ± 460	539 ± 59	39 ± 2	7.41 ± 0.01	0.2 ± 0.4
**Asphyxia**	20 ± 1	36 ± 3	2746 ± 930	2 ± 2	3598 ± 949	14,844 ± 10,406	6422 ± 1354	282 ± 106	36 ± 11	7.23 ± 0.28	−7.0 ± 4.6
**Control**	19 ± 4	39 ± 1	3492 ± 592	9 ± 1	4526 ± 838	13,596 ± 2400	8106 ± 1111	464 ± 129	41 ± 7	7.43 ± 0.04	0.8 ± 1.7
**Asphyxia + Hypothermia**	19 ± 3	39 ± 1	3128 ± 151	2 ± 1	4756 ± 266	16,658 ± 7151	5766 ± 1625	220 ± 39	47 ± 4	7.09 ± 0.25	−11 ± 4.0

Data are mean ± SD. RBC—red blood cells, WBC—white blood cells, PLT—platelets, Hct—hematocrit, BE—base deficit.

## Data Availability

The data presented in this study are available on request from the corresponding author.

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
