# Peer review of "Hypothermia after Perinatal Asphyxia Does Not Affect Genes Responsible for Amyloid Production in Neonatal Peripheral Lymphocytes"

_jcm, 2022, doi:10.3390/jcm11123263_

Round 1

Reviewer 1 Report

Please see the attached Word file.

Author Response

Reviewer#1 The manuscript by Tarkowska A et al presents a RT-PCR based study of expression levels of APP, BACE1, PSEN1, and PSEN2 in peripheral lymphocytes of neonates with perinatal asphyxia and those with perinatal asphyxia but treated with hypothermia. Authors found no significant difference between the two groups’ expression pattern or levels. ---OK.--- The control group used in this study consisted of non-healthy neonates (they were admitted to the hospital for some ailments). ---Probably the Reviewer have misunderstood the control group description. As stated in the manuscript: ‘The study included 20 neonates, 5 newborns after perinatal asphyxia plus 5 healthy neonates as a control, 5 newborns after asphyxia with hypothermic treatment, and 5 healthy neonates as a control.’ Although they had indication for taking blood for laboratory findings, none of them proved to be sick. The control group newborns had indications for diagnostic blood sample due to the following conditions: 1. infant colic (which is physiologic, occurs in healthy infants, additional examinations are performed to exclude other conditions that may cause similar symptoms) 2. breast milk jaundice – a normal condition that occurs in healthy breastfed newborns, blood sampling is performed to evaluate the bilirubin level and to exclude other conditions 3. failure to thrive due to insufficient lactation in mother – those newborns were also healthy, they only presented with poor weight gain and this problem disappeared after introducing additional milk formula feeding. The clinical signs of the control newborns were not significant - which is well documented in Table 1: Clinical characteristics of the newborns enrolled in the study.The Reviewer is probably not very familiar with neonatology and neonatal patients, that is why he thought that the control group newborns were sick, which obviously was not true.--- Sick individuals do not make a good control since their body is fighting against the ailment and this might lead to changes in the expression of several genes which might include the genes studied in this manuscript. Ideally, non-sick, healthy neonates should have been recruited for this study in order to make data comparison more robust. The whole manuscript is built around just one data-set from RT-PCR experiment done on lymphocytes obtained from neonates between 15-23 days age. Since the data is relative to the control group which itself consisted of sick neonates, reliability of this data is questionable. In addition to this, author should note that gene expression patterns is very dynamic, particularly in neonates. Therefore, for a more robust expression study, these levels should have been monitored at least on weekly basis right from when they were hospitalized for asphyxia until two-three weeks after hypothermia treatment. ---As explained above, the control group consisted of healthy neonates so comment is redundant. In any event, the decision to include infants in the control group for blood donation was made for ethical reasons. Taking blood samples from infants for research purposes only is contrary to good clinical practice and ethics. (…least on weekly basis…two-three weeks after hypothermia treatment.) And what will the legal guardians of children say, if it is scientific research in addition invasive and not treating their charges. Among the authors of the study, there are four medical doctors, including three pediatricians, we are convinced that the Polish Bioethics Committee would not consent to newborns donating blood.--- Conclusions based on the data from just one time-point RT-PCR study for a variable like gene expression which is so dynamic don’t look scientifically sound. There is no independent experimental evidence to substantiate the gene expression data. For example, Western blotting data is missing for the said proteins. ---The publication was limited to the study of gene expression at the level of transcripts due to many complex and not fully understood mechanisms regulating the transcription and translation processes of selected genes (e.g. pre-mRNA synthesis, mRNA maturation, mRNA transport to the cytoplasm, mRNA joining with ribosomes, translation , mRNA degradation, tertiary protein structure formation, post-translational modifications, protein transport, target function of proteins and protein degradation). Analyzes by many authors, incl. Gong et al. from 2017, the coexistence of mRNA-protein correlation in a single cell showed that many genes show a significant correlation between protein products and their transcripts [Gong et al. 2017]. The PubMed database contains thousands of publications of results based solely on changes in the level of transcripts, without additional testing of protein products - hence the use of real-time PCR. Due to the different half-lives of proteins in the cell, the results of Werstern blotting tests are not always reliable. Gong H, Wang X, Liu B, Boutet S, Holcomb I, Dakshinamoorthy G, Ooi A, Sanada C, Sun G, Ramakrishnan R. Single-cell protein-mRNA correlation analysis enabled by multiplexed dual-analyte co-detection Sci Rep. 2017; 7: 2776. doi: 10.1038/s41598-017-03057-5 ---The methodology used in this work (qPCR) is the same as in our previous research. Published well supported by studies in newborns with asphyxia and in an animal model of Alzheimer's disease. In this publication, we present the results of studies on a similar pattern of expression of genes related to Alzheimer's disease in newborns, but we do not write about changes in the expression of genes initiating Alzheimer's disease. The essence of this work is to identify the same trends in changes in the level of gene expression in asphyxiated neonates with identical gene expression changes in the Alzheimer's disease pathway. Of course, our theory requires furth.er research. We added a limitation to our current research at the end of the discussion.--- There is no data on the severity of asphyxia in patients and the extent of associated brain injury. For a more objective study, brain scans of both treated and untreated groups should have been done to assess the extent of injury and recovery after hypothermia treatment. In absence of brain scan data, any conclusion related to difference between untreated and hypothermia-treated subjects remain unsubstantiated. ---All newborns in the study group were diagnosed with severe or moderate perinatal asphyxia. The diagnosis of perinatal asphyxia was made on the basis of clinical, biochemical and neurological criteria, in line with current guidelines (discussed in the manuscript), including qualified newborns for hypothermia treatment. In patients after cooling down, MRI of the brain is usually performed 3-7 days after the end of the procedure. It is related to the recommendations of the health care system in Poland. So if you want to do several MRI repetitions, you have to find resources and obtain the consent of the child's legal guardians each time, which is not so easy, if the examination does not cure the child and is invasive. In contrast, transcranial ultrasound is routinely performed in every newborn with asphyxiation in the first hours of life and repeated thereafter depending on clinical indications. The effect of the cooling procedure on brain MRI results in neonates after perinatal asphyxia has already been established (Rutherford M, Ramenghi LA, Edwards AD, et al. Assessment of brain tissue injury after moderate hypothermia in neonates with hypoxic-ischaemic encephalopathy: a nested substudy of a randomised controlled trial. Lancet Neurol. 2010;9(1):39-45. doi:10.1016/S1474-4422(09)70295-9)). The overall conclusion of this research is as follows: The predictive value of MRI for later neurological impairment is not influenced by therapeutic hypothermia. We emphasize once again that the aim of our research was to assess the impact of therapeutic hypothermia on changes in gene expression related to Alzheimer's disease.--- The manuscript tries to connect elevated levels of PSEN1 and PSEN2 with subsequent hypothetical neurodegeneration similar to that observed in Alzheimer’s disease. First of all authors did not follow the neonates to see if they actually developed neurodegenerative features in their brains. Second, elevated PSEN1 and PSEN2 alone do not indicate/forecast anything about Alzheimer’s disease. Thirdly, BACE and APP levels are actually lower in asphyxia patients. This is counter-intuitive since in Alzheimer’s disease like scenario, these should have been up-regulated. Clearly, authors have over-interpreted their RT-PCR data and over-stated the conclusions. ---No one in the world knows where Alzheimer's disease begins and that it is associated with increased expression of BACE1 and APP genes. The roles of PSEN1 and PSEN2 genes are also not explained. Or perhaps the expression of the APP and BACE1 genes occurs later in survival. It should also be taken into account that some, if not all, lymphocytes have damaged APP-rich cell membranes. End-of-life remnants of lymphocytes should also be considered.---

Reviewer 2 Report

This manuscript by Tarkowska et al investigates the expression Alzheimer’s related genes in newborns following perinatal asphyxia. The findings of this project are very interesting and will be of great value to enhancing clinical studies. The study found decreased amyloid protein precusor and b-secretase, and increased presenilin 1 & 2 following asphyxia. Interestingly, therapeutic hypothermia showed similar patterns of gene expression. I note the difficulty in studying these changes in highly vulnerable population and commend the authors. The introduction is very well written and encompasses the key points to understand the rationale for this study. The authors have used a wide array of literature to make this convincing. Methodology is comprehensive, with the sample size and inclusion criteria clearly described. While the findings of the study are relatively simple, they address an important pathology that is not fully understood. They appear to support the notion that therapeutic hypothermia affords little neuroprotection to a sizeable portion of newborns, and suggests a role for these genes in production of amyloids in the brain following asphyxia.

Two queries for the authors to consider:

-Did any of the newborns present with seizures? If so, was there an observable difference in gene expression? Recent studies have shown seizures in HIE newborns may be related to BBB-disruption, and thus one might assume increased infiltration of the brain environment to these peripheral cells. Given that the cohort is quite small one would not expect further analysis to be feasible but it would be interesting to state and discuss here.

-Are the authors proposing the use of these genes as potential biomarkers? Is there an indication that more significant alterations in these genes will result in worsened neurological outcomes? Has follow-up neurobehavior been conducted on these surviving newborns?

Author Response

Reviewer#2 This manuscript by Tarkowska et al investigates the expression Alzheimer’s related genes in newborns following perinatal asphyxia. The findings of this project are very interesting and will be of great value to enhancing clinical studies. The study found decreased amyloid protein precusor and β-secretase, and increased presenilin 1 & 2 following asphyxia. Interestingly, therapeutic hypothermia showed similar patterns of gene expression. I note the difficulty in studying these changes in highly vulnerable population and commend the authors. The introduction is very well written and encompasses the key points to understand the rationale for this study. The authors have used a wide array of literature to make this convincing. Methodology is comprehensive, with the sample size and inclusion criteria clearly described. While the findings of the study are relatively simple, they address an important pathology that is not fully understood. They appear to support the notion that therapeutic hypothermia affords little neuroprotection to a sizeable portion of newborns, and suggests a role for these genes in production of amyloids in the brain following asphyxia. Thanks. Two queries for the authors to consider: -Did any of the newborns present with seizures? If so, was there an observable difference in gene expression? Recent studies have shown seizures in HIE newborns may be related to BBB-disruption, and thus one might assume increased infiltration of the brain environment to these peripheral cells. Given that the cohort is quite small one would not expect further analysis to be feasible but it would be interesting to state and discuss here. Thank you for your interesting proposal. It so happened that we did not observe any epileptic seizures in the study groups. As we plan to continue the research, it seems to be a very interesting proposition. We added a limitation to our current research at the end of the discussion. -Are the authors proposing the use of these genes as potential biomarkers? Is there an indication that more significant alterations in these genes will result in worsened neurological outcomes? Has follow-up neurobehavior been conducted on these surviving newborns? In our opinion, there is a proposal to use changes in the expression of the studied genes as potential biomarkers of the early symptoms of Alzheimer's disease. There have been no studies on the correlation of changes in the level of gene expression with changes in neurological/ neurobehavioral symptoms - they are planned in the future. We plan to observe the examined newborns for at least a few years, and preferably much longer, if possible.

Reviewer 3 Report

  1. The abstract should mention that the relative variations in AD-related genes were first made by comparing peripheral lymphocytes of non-asphyxia controls versus asphyxia newborns.
  2. This study highlights the need for more and improved therapies for an important clinical problem, perinatal asphyxia, and the study characterizes some of its molecular features for an improved examination in the future. These features are similar to other neurological disorders, suggesting a common response mechanism.
  3. The authors could request permission from Ref. 3 to add their figures 1 and 2 in this article, so their argument is complete with the full data set, and the reader does not have to search into the previous article.
  4. Discussion: In line 196-“we show for the first time changes…” – this study was not the first one; their previous study (Ref. 3) was. This experiment continued their previous work where they first reported the study on the expression of these AD-related genes in leukocytes after perinatal asphyxia.
  5. The authors should discuss that the deposition of Aβ under conditions of ischemia/hypoxia is a phenomenon that has been previously observed in neurodegenerative disorders, including ischemic stroke and brain cancer. This discussion could increase the impact and interest of the study. Work by F. Yasuno et al. (2019, Int J Geriatr Psychiatry), L. Kucheryavykh et al. (2017, Brain Res Bull) and M. Inyushin et al. (2020, Front Immunol), and others have reported accumulation of AD-related Aβ under ischemic/hypoxic conditions such as stroke and brain tumors. Most of this Aβ was shown to be contributed by platelets, as these cells are known to express large amounts of APP and produce Aβ after activation in inflammatory conditions. In AD, it has been shown that there is altered amyloid protein processing in the platelets of patients with Alzheimer's disease (RN. Rosenberg et al. (1997 Arch Neurol.)), and a similar phenomenon could also be contributing to asphyxia.
  6. Discussion: In line 196-“we show for the first time changes…” – this study was not the first; their previous study was (Ref. 3). This experiment was a continuation of their previous work, where they first reported the study on the expression of these AD-related genes in leukocytes after perinatal asphyxia.
  7. Discussion: As mentioned in line 215, the authors understand that presenilins also carry out different critical biological functions besides APP cleavage in leukocytes. These functions, such as Wnt signaling (essential for embryonic stem-cell development, tissue regeneration, cell differentiation, and immune cell regulation), regulating calcium homeostasis, and their interaction with synaptic transmission (Zhang et al., 2013, Translat Neurodegen) should be highlighted and are critical under asphyxia conditions for a newborn. Therefore, presenilin upregulation in lymphocytes may have a positive effect that remains after hypothermia.

Author Response

Reviewer #3 • The abstract should mention that the relative variations in AD-related genes were first made by comparing peripheral lymphocytes of non-asphyxia controls versus asphyxia newborns. Done. • This study highlights the need for more and improved therapies for an important clinical problem, perinatal asphyxia, and the study characterizes some of its molecular features for an improved examination in the future. These features are similar to other neurological disorders, suggesting a common response mechanism. We agree with this statement. • The authors could request permission from Ref. 3 to add their figures 1 and 2 in this article, so their argument is complete with the full data set, and the reader does not have to search into the previous article. Done. • Discussion: In line 196-“we show for the first time changes…” – this study was not the first one; their previous study (Ref. 3) was. This experiment continued their previous work where they first reported the study on the expression of these AD-related genes in leukocytes after perinatal asphyxia. Done. • The authors should discuss that the deposition of Aβ under conditions of ischemia/hypoxia is a phenomenon that has been previously observed in neurodegenerative disorders, including ischemic stroke and brain cancer. This discussion could increase the impact and interest of the study. Work by F. Yasuno et al. (2019, Int J Geriatr Psychiatry), L. Kucheryavykh et al. (2017, Brain Res Bull) and M. Inyushin et al. (2020, Front Immunol), and others have reported accumulation of AD-related Aβ under ischemic/hypoxic conditions such as stroke and brain tumors. Most of this Aβ was shown to be contributed by platelets, as these cells are known to express large amounts of APP and produce Aβ after activation in inflammatory conditions. In AD, it has been shown that there is altered amyloid protein processing in the platelets of patients with Alzheimer's disease (RN. Rosenberg et al. (1997 Arch Neurol.), and a similar phenomenon could also be contributing to asphyxia. Thank you very much for a very valuable indication of complementing the discussion. Done. • Discussion: In line 196-“we show for the first time changes…” – this study was not the first; their previous study was (Ref. 3). This experiment was a continuation of their previous work, where they first reported the study on the expression of these AD-related genes in leukocytes after perinatal asphyxia. Done. • Discussion: As mentioned in line 215, the authors understand that presenilins also carry out different critical biological functions besides APP cleavage in leukocytes. These functions, such as Wnt signaling (essential for embryonic stem-cell development, tissue regeneration, cell differentiation, and immune cell regulation), regulating calcium homeostasis, and their interaction with synaptic transmission (Zhang et al., 2013, Translat Neurodegen) should be highlighted and are critical under asphyxia conditions for a newborn. Therefore, presenilin upregulation in lymphocytes may have a positive effect that remains after hypothermia. Thank you very much once again for a very valuable tip that complements the discussion. Done.

Round 2

Reviewer 1 Report

Authors have adequately addressed all the concerns raised in the previous round of review. I appreciate authors for their time and effort in providing point-by-point response to the criticism.

Author Response

We would like to thank the reviewer number 1 for explaining unclear issues to each other. The cooperation was very fruitful and to the point.

Reviewer 3 Report

1.Figures 1 and 2 must state the following at the end of the figure legend: *Reprint or Data from: “Title of article”  by First Author et al., Year, Journal, Vol., page. For specific instructions go to:  (https://www.mdpi.com/authors/rights).

2. Please check for orthographical mistakes, for example, in line 287- Please correct “amyloidd” change to ‘amyloid’, line 389 must be corrected, in general, the authors must do a checkup for orthographical and semantic mistakes throughout the manuscript.

3. It would be interesting to follow up on the development of AD markers as these infants grow old.

Author Response

Reviewer #3. 1.Figures 1 and 2 must state the following at the end of the figure legend: *Reprint or Data from: “Title of article” by First Author et al., Year, Journal, Vol., page. For specific instructions go to: (https://www.mdpi.com/authors/rights).

Done. 2.

Please check for orthographical mistakes, for example, in line 287- Please correct “amyloidd” change to ‘amyloid’, line 389 must be corrected, in general, the authors must do a checkup for orthographical and semantic mistakes throughout the manuscript.

Done.

3. It would be interesting to follow up on the development of AD markers as these infants grow old.

We think about it.